# GP-SWAT (v1.0): A two-level graph-based parallel simulation framework for the SWAT model

Dejian Zhang[b,c,★], Bingqing Lin[a,★], Jiefeng Wu[d], Qiaoying Lin[a,#]

[a]Department of Resources and Environmental Sciences, Quanzhou Normal University, Donghai Street 398, Quanzhou, Fujian 362000, China
[b]College of Computer and Information Engineering, Xiamen University of Technology, Ligong Road 600, Xiamen, Fujian 361024, China
[c]Digital Fujian Institute of Big Data for Natural Hazards Monitor, Ligong Road 600, Xiamen, Fujian 361024, China
[d]School of Hydrology and Water Resources, Nanjing University of Information Science and Technology, Nanjing 210000, China

[#]*Correspondence to*: Qiaoying Lin (qylin@qztc.edu.cn)

★ These authors contributed equally to this work.

**Abstract.** High-fidelity and large-scale hydrological models are increasingly used to investigate the impacts of human activities and climate change on water availability and quality. However, the detailed representations of real-world systems and processes contained in these models inevitably lead to prohibitively high execution times, ranging from minutes to days. This becomes computationally prohibitive or even infeasible when large iterative model simulations are involved. In this study, we propose a generic two-level model parallelization scheme to reduce the run time of computationally expensive model applications through a combination of model spatial decomposition and the graph-parallel Pregel algorithm. Taking the Soil and Water Assessment Tool (SWAT) as an example, we implemented a generic tool named GP-SWAT, enabling model-level and subbasin-level model parallelization on a Spark computer cluster. We then evaluated GP-SWAT in two sets of experiments to demonstrate the potential of GP-SWAT to accelerate single and iterative model simulations and to run in different environments. In each test set, GP-SWAT was applied for the parallel simulation of eight synthetic hydrological models with different input/output (I/O) burdens and river network characteristics. The single-model parallelization results showed that GP-SWAT can achieve a 1.8-5.8-times speedup. For multiple simulations with subbasin-level parallelization, GP-SWAT yielded a remarkable speedup of 6.62-27.03 times. In both cases, the speedup ratios increased with increasing model complexity. The experimental results indicate that GP-SWAT can effectively solve high-computational-demand problems of the SWAT model. In addition, as a scalable and flexible tool, it can be run in diverse environments, from a commodity computer running the Microsoft Windows operating system to a Spark cluster consisting of a large number of computational nodes. Moreover, it is possible to apply this generic scheme to other subbasin-based hydrological models or even acyclic models in other domains to alleviate input/output (I/O) demands and optimize model computational performance.

## 1 Introduction

With the enhanced availability of high-resolution remote sensing data and long periods of hydrometeorological data, hydrologists are increasingly building high-fidelity hydrological models to investigate water availability (Liang et al., 2020), water quality (Fang et al., 2020), climate change (Cai et al., 2016), and watershed management options (Jayakody et al., 2014;Qi and Altinakar, 2011;Lee et al., 2010). However, these hydrological models, which contain detailed representations of real-world systems and processes, can demand large computational budgets and require prohibitively high execution times, ranging from minutes to days (Razavi et al., 2010). Because modeling practices such as model calibration and uncertainty analysis usually involve thousands of model evaluations or more, they may sometimes become computationally prohibitive or even infeasible (Razavi and Tolson, 2013). Thus, the effective use of computationally expensive simulations remains a challenge for many applications involving a large number of model simulations.

In general, there are four broad research methods for alleviating the computational burden associated with computationally expensive model applications: (1) utilizing metamodeling approaches (Chandra et al., 2020;Sun et al., 2015), (2) developing computationally efficient algorithms (Humphrey et al., 2012;Joseph and Guillaume, 2013), (3) opportunistically avoiding model evaluations (Razavi et al., 2010), and (4) utilizing parallel computing technologies and infrastructures (Yang et al., 2020;Huang et al., 2019;Wu et al., 2013;Wu et al., 2014;Zamani et al., 2020). The first three methods share the same goal of reducing the computational demand by using lightweight surrogate models, decreasing the number of model simulations, and terminating model execution early when the simulation result is poorer than expected. The fourth method adopts a different strategy of boosting model application performance by optimizing the efficiency of computational resource utilization. These four strategies are largely complementary, and in practice, a given modeling approach may employ a combination of two or more of these strategies (e.g., coupling parallelization with a computationally efficient optimization algorithm). In this study, we propose a scheme in which the last strategy is adopted to optimize the efficiency of generic modeling activities. The remainder of this section will briefly review model-level, submodel-level, spatial-decomposition parallelization methods and the effects of parallel computing techniques and resources on model performance to establish the context for the research presented in this paper.

Model parallelization can be achieved at the model and/or submodel level. Model-level parallelization is applicable for modeling routines such as model calibration, sensitivity and uncertainty analysis, and identifying beneficial management practices, which involve a large number of model simulations; however, it cannot be used to reduce the run time in circumstances involving only one model simulation (e.g., flood forecasting). There are many model-level model parallelization methods in the literature that were utilized to boost the performance of modeling practices. These applications include but are not limited to model calibration (Zhang et al., 2013;Ercan et al., 2014;Gorgan et al., 2012), sensitivity analysis (Khalid et al., 2016;Hu et al., 2015), uncertainty analysis (Zhang et al., 2016;Wu and Liu, 2012;Zamani et al., 2020) and beneficial management practices recognition (Liu et al., 2013).

By contrast, submodel-level parallelization is effective in cases involving a single model simulation, and it can also be applied in combination with model-level parallelization to maximize the performance of modeling routines involving a large number of model simulations. For example, Wu et al. (2013) improved the performance of the SWAT model by distributing subbasin simulations to different computational cores through the MPI. Wang et al. (2013) developed the temporal-spatial discretization method (TSDM), in which the parallelization degree of subbasins was exploited to the maximum extent by properly organizing the simulation sequences of dependent subbasins. Liu et al. (2014) proposed a layered approach with Open Multi-Processing (OpenMP) to boost the performance of the fully sequential dependent hydrological model (FSDHM) in a multicore computer. This work was further extended with the Message Passing Interface (MPI) to achieve subbasin-level model parallelization in a computer cluster (Liu et al., 2016). Based on the MPI and OpenMP frameworks, Zhu et al. (2019) introduced the spatially explicit integrated modeling system (SEIMS) to achieve submodel parallelization in a computer cluster consisting of multiple nodes. However, submodel-level parallelization usually requires model reconstruction to enable the parallel simulation of model components and to achieve the communication among the components that is necessary for integrating the model results. As a result, a steep learning curve is expected for modelers who are unfamiliar with the model source codes. Although there are some parallel computation frameworks available that can facilitate model parallelization at the submodel level, such as Open MPI and the OpenMP application programming interface (API), it is still a very tedious and time-consuming process.

For spatially explicit acyclic models, it is possible to decompose a large-scale model into multiple smaller models and properly orchestrate the simulation processes of these smaller models to generate integrated results for the original model. This spatial decomposition and merging strategy has all of the advantages of submodel-level parallelization but requires no model reconstruction. Taking the SWAT model as an example, a large-scale watershed model involving multiple subbasins can be split into multiple smaller models, each of which consists of only one subbasin (hereafter referred to as subbasin models). The stream flow and chemical loadings from upstream subbasins can be treated as boundary conditions, which can be incorporated as point sources. Through proper organization of the simulation of these models, a result identical to that of the original model can be achieved. In this strategy, upstream subbasin models must be simulated before downstream subbasin models; however, sibling subbasin models can be simulated in parallel to optimize the model performance (for detailed information about the implementation of subbasin-level parallelization for SWAT model, readers are referred to Yalew et al. (2013)).

In addition, the choice of parallel computing techniques and resources has a great influence on model parallelization performance. Recent studies have demonstrated the feasibility of combining parallel computation techniques with shared-memory or distributed-memory systems to enhance the efficiency of model parallelization. Examples of model parallelization with shared-memory systems include the works of Rouholahnejad et al. (2012), Wu and Liu (2012), and Joseph and Guillaume (2013). However, the performance gains of these tools or methods are hindered by the poor scalability of shared-memory systems. On the other hand, distributed-memory model parallelization systems offer better scalability and, thus, better performance. A distributed-memory model parallelization system in the form of grid, cluster or cloud computing

allows computational nodes to be dynamically added to boost model parallelization performance. For example, Whittaker (2004) and Confesor Jr and Whittaker (2007) presented a parallel method by combining cluster-based parallel computing with the nondominated sorting genetic algorithm version II on a Beowulf cluster consisting of a server and 12 computation nodes to facilitate the analysis of uncertainty in the Soil and Water Assessment Tool (SWAT). Zhang et al. (2013) established a Python-based parallel computing package called PP-SWAT by combining Python, the Message Passing Interface (MPI) standard for Python and the open-source MPI standard Open MPI for the multiple-objective calibration of SWAT. Gorgan et al. (2012) implemented the gSWAT application by leveraging grid computing technologies and infrastructures for the calibration of extensive hydrological models. However, these methods have one major limitation: modelers must use a parallel framework (e.g., Open MPI) to cope with model communications, failover, task management, etc. Recently, the growth of cloud computing systems and big data techniques has enabled modelers to avoid the aforementioned limitations. For example, with cloud computing, users can easily build their own private clouds or simply use third-party facilities (e.g., Azure HDInsight, Amazon Web Services EMR, or Google Dataproc). In addition, parallel frameworks for processing big data, such as Hadoop, Spark and Flink, are now available to alleviate the burden placed on modelers for addressing low-level programming tasks such as communications, failover, and task management. These advantages have sparked research on efficient solutions to complex high-dimensional computational problems. For example, Humphrey et al. (2012) and Ercan et al. (2014) established a calibration system for SWAT based on Microsoft Windows Azure and the dynamically dimensioned search method and achieved a significant speedup in SWAT calibration. Hu et al. (2015) utilized Hadoop-based cloud computing techniques and a variance decomposition approach based on polynomial chaos expansion for the improvement of global sensitivity analysis with large-scale socio-hydrological models. Zhang et al. (2016) implemented the cloud-based Calibration and Uncertainty analysis Tool for SWAT (CUT-SWAT) using Hadoop and generalized likelihood uncertainty estimation method.

In this study, we propose a two-level model parallelization scheme based on a combination of the graph-parallel Pregel algorithm and model spatial domain decomposition. The objective of this study is to create a simulation-accelerated tool for the SWAT model by adopting both watershed-level and subbasin-level parallelization, without the model reconstruction to achieve data communication among model components that current solutions (e.g., MPI) usually require. We hope that this tool will help IT practitioners or modelers improve model performance without requiring specific domain knowledge of the hydrological model. In accordance with this scheme and goal, a graph-parallel simulation tool for SWAT (named GP-SWAT) has been developed using an open-source general-purpose distributed cluster computing framework, Spark. GP-SWAT has been assessed in two sets of experiments to demonstrate its potential to accelerate single and iterative model simulations at different parallelization granularities when implemented on a computer running the Windows operating system (OS) and on a Spark cluster consisting of five computational nodes. Experiment set one was conducted to illustrate that GP-SWAT can be used to perform subbasin-level model parallelization using a multicore computer running the Windows OS, while in experiment set two, GP-SWAT was assessed for iterative model runs. For each experiment in the latter set, subbasin- and watershed-level parallelization schemes were employed to execute 1000 model simulations with one to five parallel tasks

implemented on each computational node. In each of the test cases, GP-SWAT was evaluated based on eight synthetic hydrological models representing different input/output (I/O) burdens and different levels of river network complexity.

## 2 Materials and methods

### 2.1 Graph representation of hydrological models

A property graph is a directed graph with properties attached to each vertex and edge. Each vertex is keyed by a unique identifier. Similarly, each edge has corresponding source and destination vertex identifiers. In many domains of the natural and social sciences, relationships of interest can be described by property graphs, such as the relationships in traffic and social networks. Accordingly, many graph algorithms have been devised to simplify and improve the performance on related analytical tasks, including methods for the parallel processing of graph-based data. From this perspective, the relationships between the components of a distributed hydrological model can be described by a dendriform property graph. Figure 1a demonstrates how a simple watershed model can be represented with a property graph. Each subbasin can be represented as a vertex with an identifier and two properties denoting how many subbasins exist directly upstream of the current subbasin (referred to as subNo hereafter) and the number of these directly upstream subbasins for which the simulation process has been completed in the current computation step (referred to as finSubNo hereafter). Each edge denotes an upstream-downstream flow drainage relationship. Modeling routines involving iterative simulations can also be represented in the form of property graphs. In this case, each simulation is represented by a subgraph of the integrated graph, and each outlet vertex of these subgraphs is connected to a virtual vertex to form the integrated graph (figure 1b). To uniquely identify a vertex in the integrated graph, the vertex identifier consists of both the subbasin number and the simulation number. In addition, the virtual vertex is identified by a special number (i.e., -1 in our case). Such an integrated property graph representing a modeling routine involving iterative simulation can also be interpreted as a property graph representing a large-scale virtual hydrological model consisting of many identical landscapes (represented by the same model with different parameters) that are connected to a virtual outlet. Moreover, in this case, each vertex can be considered to represent a subbasin model or watershed model, and thus, it is possible to achieve model parallelization at the watershed and subbasin levels (figure 1c).

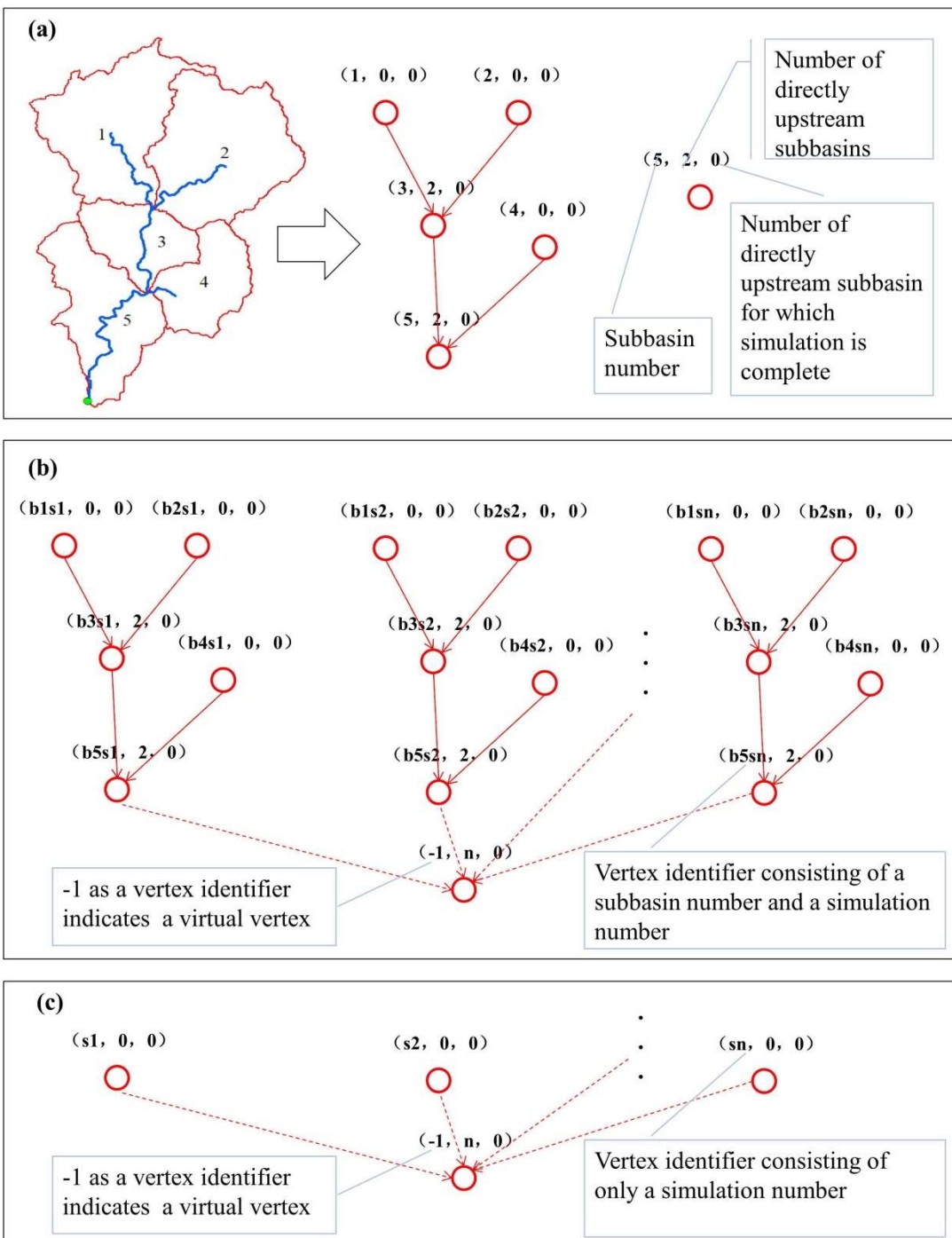

**Figure 1: Using directed acyclic graphs to represent watershed route information (a), and iterative simulations at the subbasin level (b) and model level (c).**

## 2.2 Design and implementation of GP-SWAT

Apache Spark is an open-source general-purpose distributed cluster computing framework that provides distributed task dispatching, scheduling, and graph functionalities through an API available in the Java, Python, Scala, and R languages. Apache Spark can run in diverse environments, ranging from a single computer (running the Microsoft Windows or Linux OS) to a computer cluster consisting of a large number of computational nodes hosted in an in-house or cloud environment.

GP-SWAT is designed to work with the Spark graph API for model-level and subbasin-level parallelization. Figure 2 outlines the main components and required environments of GP-SWAT and their interactions. The required environments include a shared storage environment for sharing model simulation results and other information and a Spark cluster for the parallel simulation of models. In this study, the network file system (NFS) protocol was used as the basis for the shared storage to exchange data among the executors in the computer cluster. GP-SWAT consists of two components: a

preprocessing program and a driver program. The preprocessing program is used to generate a calibration parameter set, extract route information from the watershed configuration file (fig.fig) for the SWAT model, and create a watershed configuration file for each subbasin model. The second component, the driver program, creates a hydrological model property graph from the route information generated by the preprocessing program and defines how the models are to be simulated in parallel; its output is then replicated to the executors to actually carry out model parallelization.

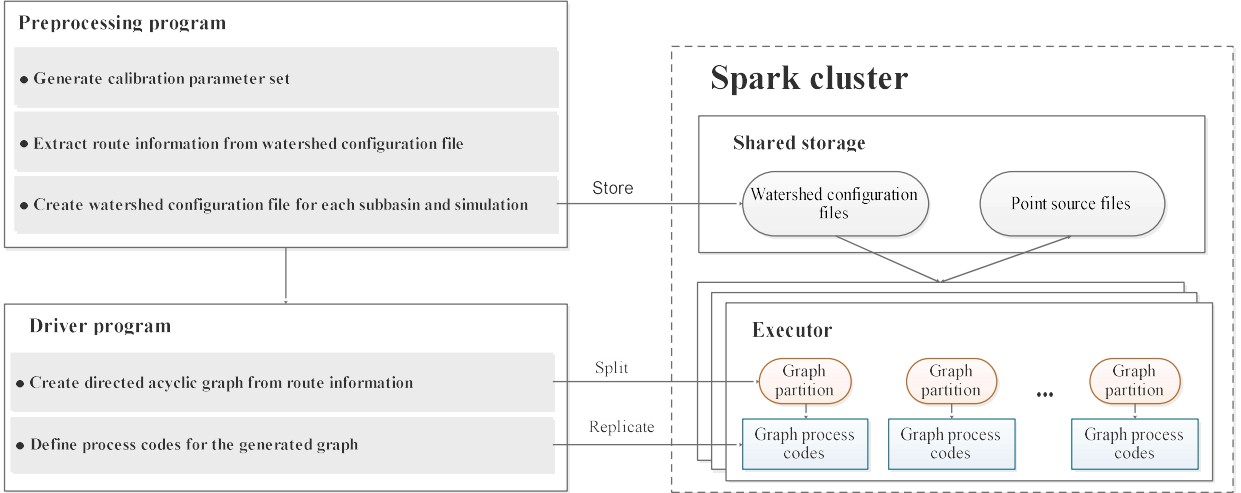

**Figure 2: Schematic diagram of GP-SWAT.**

Figure 3 shows the code of the driver program as implemented in Scala. The code for creating the hydrological model property graph (lines 1-12 in figure 3) is straightforward. Vertices and edges are first created based on the route information extracted from the watershed configuration file by the preprocessing program. The property graph for the hydrological

model is then created by taking these vertices and edges as function arguments. Model parallelization is achieved by means of the graph-parallel Pregel algorithm (lines 14-33 in figure 3). The Pregel algorithm is a message-based algorithm. It consists of a series of supersteps in which each vertex receives the sum of its inbound messages from the previous superstep,

the vertex properties are updated in accordance with the merged messages, and messages are then sent to neighboring vertices in the next superstep. The Pregel algorithm terminates when there are no messages remaining or the maximum number of iterations defined by the developers has been reached. The Pregel algorithm as implemented in Spark takes two argument lists. The first argument list consists of the initial message, the maximum number of iterations, and the edge direction in which to send messages. The second argument list consists of the user-defined functions for receiving messages (receiveMsg program), computing messages (sendMsg program), and combining messages (mergeMsg program). To achieve model parallelization with the Pregel algorithm, "0" is specified as the initial message, the maximum number of iterations is assigned to be a very large number (Int.MaxValue) to ensure that the Pregel algorithm will terminate only once no messages remain, and the edge direction is set to "EdgeDirection.Out", indicating that messages are sent only to downstream neighboring vertices. Three anonymous functions are provided to receive and process messages (lines 16-22 in figure 3), generate messages for the next superstep (lines 24-30 in figure 3), and merge messages (line 32 in figure 3). In the first anonymous function, finSubNo (the second property of a vertex, denoting the number of directly upstream subbasin models for which simulation has been completed) is updated by adding the merged message (denoting the number of directly upstream subbasin models for which simulation had been completed in the previous superstep), and finSubNo is then compared with subNo (denoting the number of subbasins directly upstream of the current subbasin). If finSubNo is equal to subNo (meaning that simulation is complete for all directly upstream subbasins), then the subbasin model represented by the current vertex is executed through an external function that will be discussed later. In the second anonymous function, used to compute the messages for the next superstep, a value of "1" is sent to the downstream neighboring vertex if the current subbasin model has been simulated in the current superstep; otherwise, a value of "0" is sent, indicating that no subbasin model was executed. The third anonymous function simply adds two messages from upstream vertices and returns the result of the addition operation (this function is invoked n-1 times, where n is the number of inbound messages).

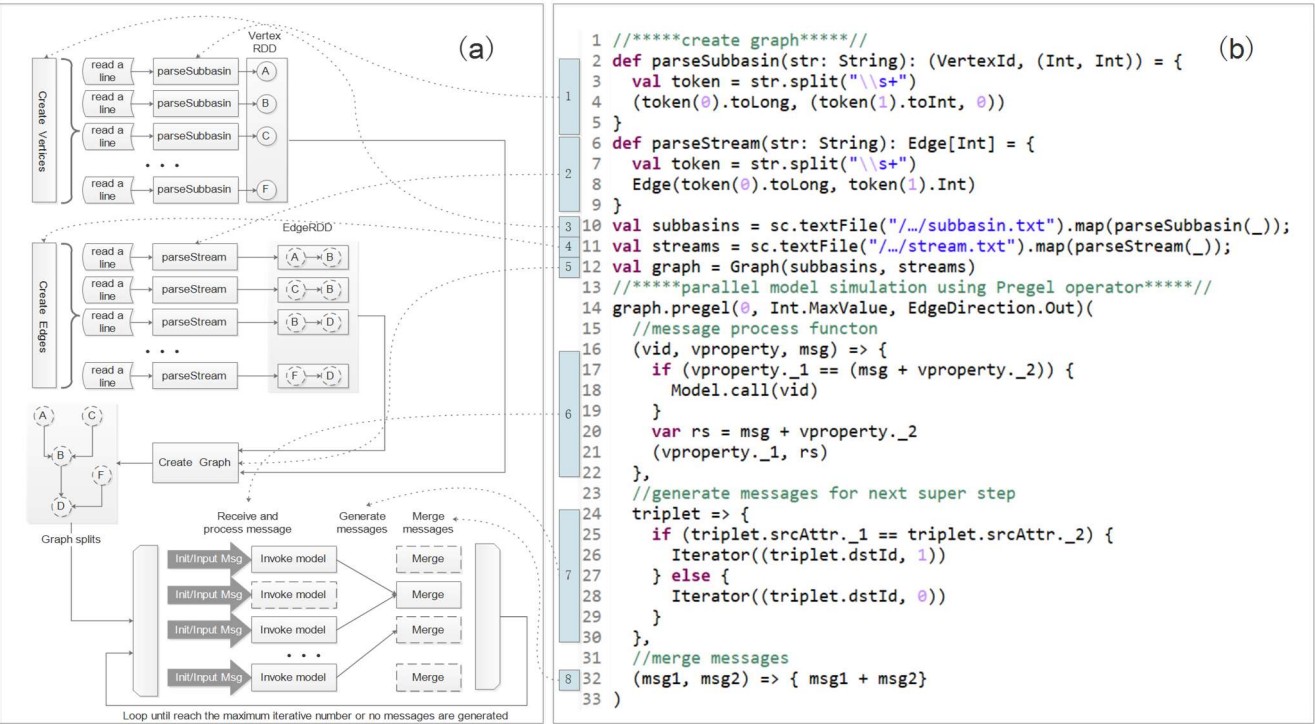

**Figure 3: Schematic diagram of model parallelization with the Pregel algorithm (a) and code snippet of the driver program (b) (note that imported packages and some signatures have been removed for simplicity).**

In figure 4, we demonstrate the model parallelization of a simple hydrological model consisting of five subbasins (figure 1a) using the Pregel algorithm. In the initial superstep, the receiveMsg program is invoked on all vertices and is passed the default message "0". As seen from figure 4, finSubNo is equal to subNo at vertices 1, 2 and 4; thus, these three subbasin models are executed. Next, the sendMsg program is invoked on all vertices that have inbound messages and downstream neighboring vertices. For vertices 1, 2 and 4, at which subbasin model simulation has been performed, "1" is sent to their downstream neighboring vertices. Because vertex 3 did not execute simulation of its subbasin model, "0" is sent. Finally, at the end of each superstep, the mergeMsg program is invoked on vertices with at least two inbound messages in the next superstep. In the second superstep, the receiveMsg program is invoked on vertices 3 and 5, which have inbound messages. Because finSubNo is equal to subNo only at vertex 3, only the subbasin model associated with this vertex is executed, and sendMsg sends "1" to its downstream neighboring vertex 5. In superstep 3, the receiveMsg program is invoked on vertex 5 and triggers the simulation of its associated subbasin model, as its finSubNo value has reached the number of directly upstream subbasins.

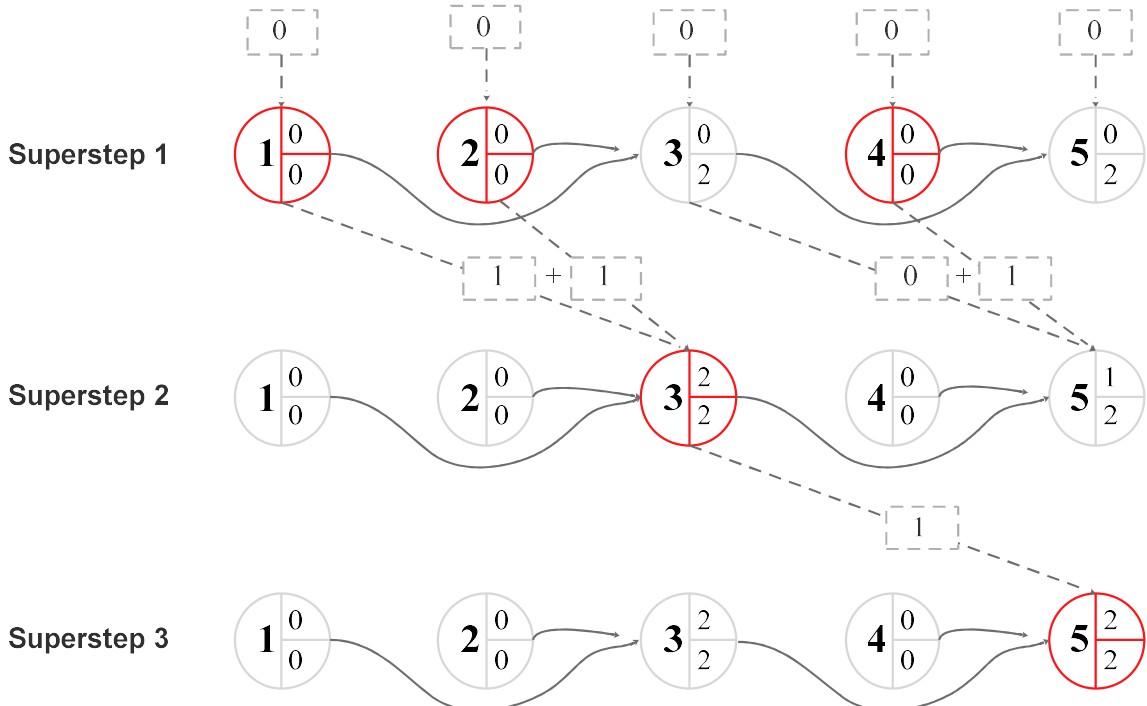

**Figure 4: Demonstration of the parallel simulation of subbasin models of the example watershed (Figure 1a) using the Pregel operator of GraphX (the number in the left part of a vertex is the subbasin number, the lower right number denotes how many upstream subbasins are directly connected to the current subbasin, the upper right number indicates the number of directly upstream subbasins for which simulation has been completed, and a red vertex denotes that the corresponding subbasin model is simulated in that superstep).**

The function for invoking a model (Model.call in figure 3) is implemented in Java to reuse components that have been implemented in previous studies (Zhang et al., 2016). Because the models are designed to be executed in parallel within a computational node and among nodes, we must ensure that a model can be executed only once at a given time. Thus, the first step of model simulation is to find a model that is not occupied; this task is achieved through a combination of the synchronous mechanism and static indicators of Java. When a free model is identified, the watershed configuration file is replaced, and the upstream point-source files stored in the NFS shared storage are copied to the model directory. Next, the subbasin model inputs are edited in accordance with the parameter set of the current simulation. Finally, the model is executed, and the model output is copied to the NFS shared storage for later use.

**3 Case study**

In this study, synthetic hydrological models of the Harp Lake and Jinjiang watersheds (figure 5) were used to evaluate the performance of GP-SWAT. The Harp Lake and Jinjiang watersheds have distinct characteristics in terms of spatial extents

and stream network complexities and were thus used as representatives of simple and complex watersheds, respectively, to demonstrate how watershed characteristics can affect the GP-SWAT performance. The Harp Lake catchment, which covers a drainage area of 5.20 km², is located within the Boreal Shield ecozone in south-central Ontario, Canada. It has six streams that drain directly into Harp Lake. The land uses include mixed deciduous and coniferous forest (>75% of the total land area), lakes (about 15%), and wetlands and ponds (about 7%). The annual mean temperature ranges from 3.5 °C to 6.5 °C, with an average of 4.9 °C. The mean January and July air temperatures are -10.5 °C and 18.4 °C, respectively. Jinjiang catchment is located in south-eastern Fujian Province of China. It has a drainage area of 5629 km², which is occupied predominantly by mountains and rangelands. There are two major river branches within the Jinjiang River, i.e., the east branch and the west branch, which merge into the main stream 2.5 km upstream of the Shilong gauge station. With a typical humid sub-tropical climate, the area has an annual mean temperature of 20 °C and average annual precipitation of 1686 mm.

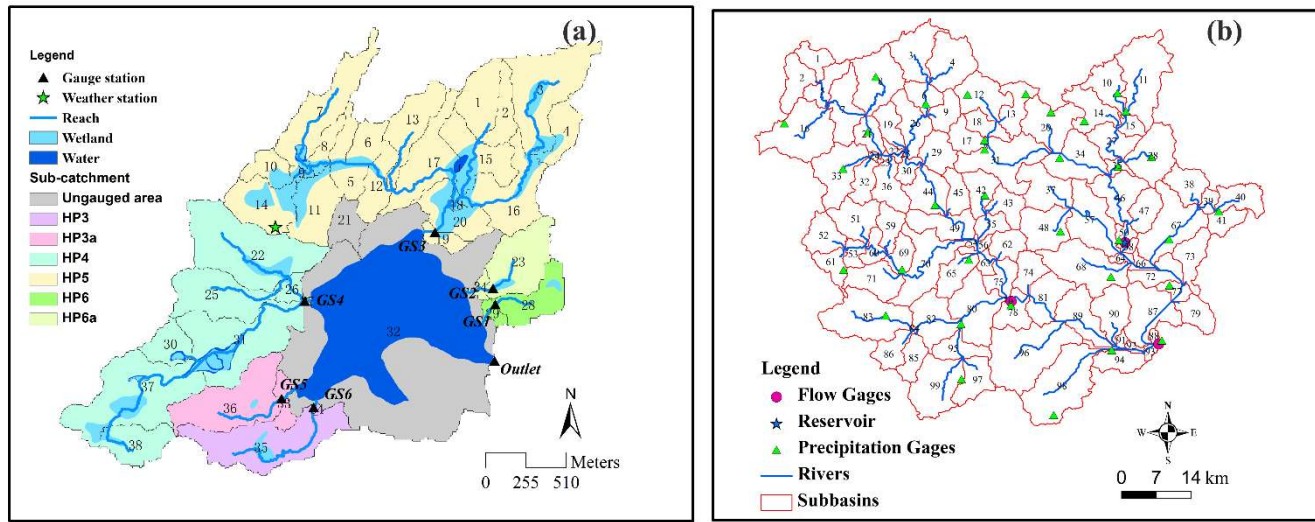

**Figure 5: Watershed delineations for the Harp Lake (a) and Jinjiang (b) watersheds.**

Eight synthetic hydrological models representing different I/O burdens and different levels of river network complexity were used to evaluate the efficiency of GP-SWAT. These synthetic models were built based on the Harp Lake and Jinjiang hydrological models (Fu et al., 2014;Zhang et al., 2020;Zhang et al., 2015). These synthetic models were created with a tool implemented in Java Program Language (available at https://github.com/djzhang80/models-and-tools/createproject.jar). This tool allows users to create synthetic models with any desired HRUs per subbasin for performance evaluation but not to create meaningful hydrological models. The Harp Lake and Jinjiang hydrological models include 38 and 99 subbasins, respectively. For the various synthetic hydrological models, the number of subbasins, the simulation period and other configuration parameters are the same, but different numbers of hydrological response units (HRUs) in the subbasins are considered. The

synthetic models based on the Jinjiang hydrological model (JJ) with 5, 50, 100 and 150 HRUs per subbasin are denoted by JJ1, JJ2, JJ3 and JJ4, respectively. The synthetic models based on the Harp Lake hydrological model (HP) with 5, 50, 100 and 150 HRUs per subbasin are denoted by HP1, HP2, HP3 and HP4, respectively.

Two sets of experiments were established to verify the use cases of GP-SWAT: a) it can be used to accelerate single model simulations on a multicore computer running Microsoft Windows through subbasin-level parallelization, and 2) it can also be employed to run parallel model simulations on a Spark computer cluster through both model-level and subbasin-level parallelization. In this study, the performance of GP-SWAT was evaluated based on the speedup metric, which is defined as follows:

$$Speed_{act} = (ST * n)/JET, \tag{1}$$

where $n$ is the total number of simulations for a given job, $JET$ is the total execution time of the job when run in the test environment, $ST$ is the average execution time of one model simulation, and $Speedup_{act}$ measures how much faster a program runs in parallel compared to being run sequentially on a single computer. For subbasin-level parallelization, we also calculated the theoretical speedup, which considers the theoretical speedup that a model can achieve under different test configurations. The theoretical speedup is calculated as follows:

$$Speed_{ref} = Sub_{num}/\sum_{i=1}^{n} Ceil(Count_i/PT_{num}), \tag{2}$$

where $sub_{num}$ is the number subbasins of the hydrological model under test, n is the total number of supersteps, $i$ denotes the $i$-th superstep, $Count_i$ is the number of subbasin models simulated in the $i$-th superstep, $PT_{num}$ is the number of parallel tasks performed at an executor, and $Ceil$ is a function that returns the smallest integer value that is greater than or equal to a predetermined parameter value.

Experiment set one was carried out on a workstation with 24 processors operating at a frequency of 2.67 GHz, 24 GB of RAM, and 1 TB of disk storage, running the Windows 2012 Server OS. Spark was configured with one executor, and a maximum of 24 cores were allowed in this executor. The eight synthetic hydrological models were used to evaluate subbasin-level parallelization with the number of parallel executor tasks ranging from one to 24. Experiment set two was carried out on a Spark cluster consisting of 5 computational nodes. Each computational node had a quad-core CPU, 8 GB of RAM and 50 GB of disk storage. The CPU frequency was 2.2 GHz, and each node was running 64-bit Linux as the OS. In this Spark cluster, each node could have 1-5 executors, but each executor was allowed to have only one core. Model-level and subbasin-level parallelization of the eight synthetic hydrological models was conducted on this Spark cluster with 5, 10, 15, 20 and 25 executors.

## 4 Results and discussion

### 4.1 Performance analysis

Spark applications can run either in local mode (nondistributed deployment mode with a single Java Virtual Machine (JVM)) or in cluster mode. When run in local mode, Spark can be deployed on a computer with a Microsoft Windows, Mac, or

Linux OS. Experiment set one was designed to verify that GP-SWAT can be used to perform subbasin-level model parallelization using a multicore computer running the Windows OS. Each synthetic model was simulated 10 times with 1-24 cores, and the actual speedup values were calculated using the average execution time. Figure 6 shows the actual and theoretical speedups versus the number of parallel tasks performed in local mode. In general, the actual speedup values

increase with increasing model complexity for both the HP- and JJ-based synthetic models, indicating that subbasin-level parallelization works especially well for complex models. For less complex models, such as HP1 and JJ1, the best speedup values are 1.8 and 2.3, respectively. However, for more complex models, a much better speedup can be achieved. For example, the maximum speedup values achieved for HP4 and JJ4 are 4.8 and 5.8, respectively. It is noted that the actual speedup values surpass the theoretical values for HP3, HP4, JJ3 and JJ4. The reason of this phenomenon is that the speedup

is achieved not only by simulating the model components in parallel but also by alleviating the model's IO burden. For example, the execution time of the undivided model of JJ4 is 425.02 s, but the execution time of its subbasin model is only 2.35 s. In other words, even if the subbasin models of JJ4 are executed in sequence, this still can lead to a reduction of 192.37 s (425.02-2.35*99). This is because the original SWAT model reads all model inputs at once in the initial stages of model execution, which can easily cause I/O saturation and thus prolong the model execution time. The theoretical speedup

ratios were calculated against the execution time of the original model. In contrast to the theoretical speedup ratio calculation, the actual speedup ratios were calculated by considering both performance gains from subbasin-level parallelization and IO alleviation. Figure 7 shows the job execution times of these synthetic models versus the parallel tasks performed on a Windows server. The job execution time includes the times associated with job initiation, task orchestration and model result transfer (system time) and the time pertaining to model execution (model time). The model execution times were calculated

by summing the total execution time for each subbasin model, and the system execution times were calculated by subtracting the model execution times from the job execution times. As shown in Figure 7, the system execution times were stable for both the HP-based and JJ-based models with different HRUs.

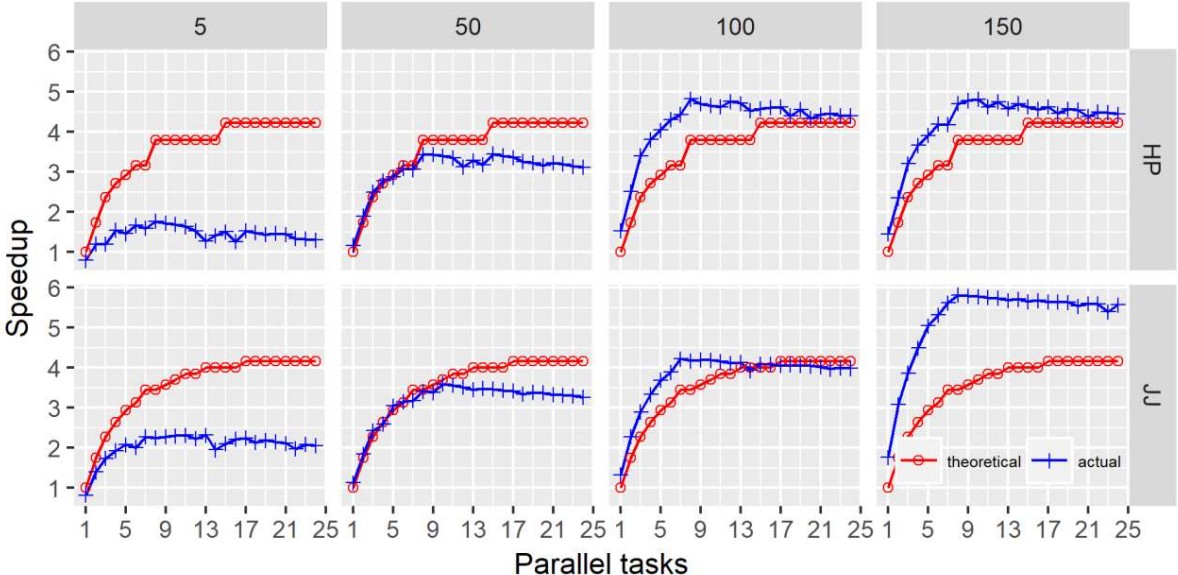

**Figure 6: Speedup results achieved for single model simulation on a Windows server versus the number of parallel tasks for the eight synthetic hydrological models.**

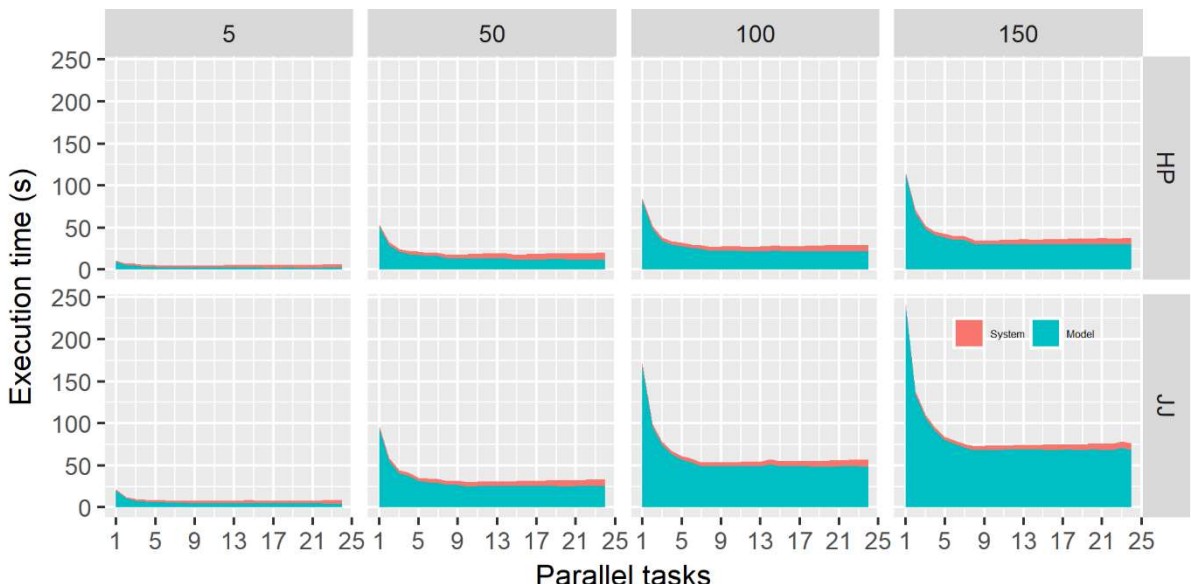

**Figure 7: Job execution times on a Windows server versus the number of parallel tasks for the eight synthetic hydrological models.**

In experiment set two, we evaluated the performance of GP-SWAT for iterative model runs. For each experiment in this set, subbasin- and watershed-level parallelization (hereafter referred to as the decomposed mode and the integrated mode,

respectively) were employed to execute 1000 model simulations in the test environment with one to five parallel tasks implemented at each computational node. The speedup values achieved in the integrated mode (red) and in the decomposed mode (blue) are plotted against the number of parallel tasks in figure 8. For simple synthetic models (i.e., HP1 and JJ1), the speedups in the integrated mode surpass those in the decomposed model, while for more complex models (models other than HP1 and JJ1), the speedups in the decomposed mode are better than those in the integrated mode. This is a result of the conflict between the system overheads (including job initiation, task orchestration, and model result transfer) and performance gains associated with subbasin-level parallelization. In general, the speedup gradually increases with up to 20 parallel tasks and then slightly decreases as the number of parallel tasks continues to increase. Similar to the case of single model runs, we also observe that the speedup increases with increasing model complexity for both the decomposed and integrated modes. As discussed before, this phenomenon is caused by the additional performance gains resulting from model decomposition. For example, the maximum speedups for the least complex models HP1 and JJ1 are 6.62 and 8.34, respectively, while the speedups for HP4 and JJ4 are 22.75 and 27.03, respectively. Figure 9 shows the execution times against the number of parallel tasks. In general, the fraction of system time decreases with increasing parallel tasks and increases again when the number of parallel tasks reaches 4 and/or 5. We believe that larger system times with smaller parallel tasks occur because the model execution overheads are relatively small, while in cases with large parallel tasks, the conflict of demanding resources certainly leads to a longer time required for performing task dispatch and securing computational resources.

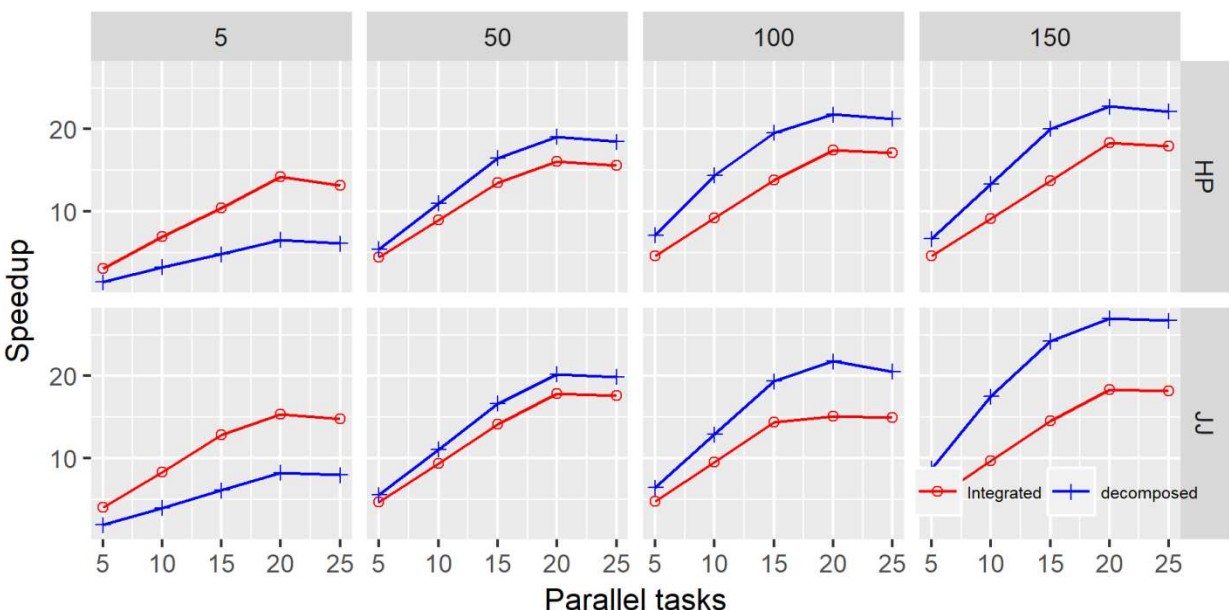

**Figure 8: Speedup results achieved for iterative model simulations on a Spark cluster versus the number of parallel tasks for the eight synthetic hydrological models.**

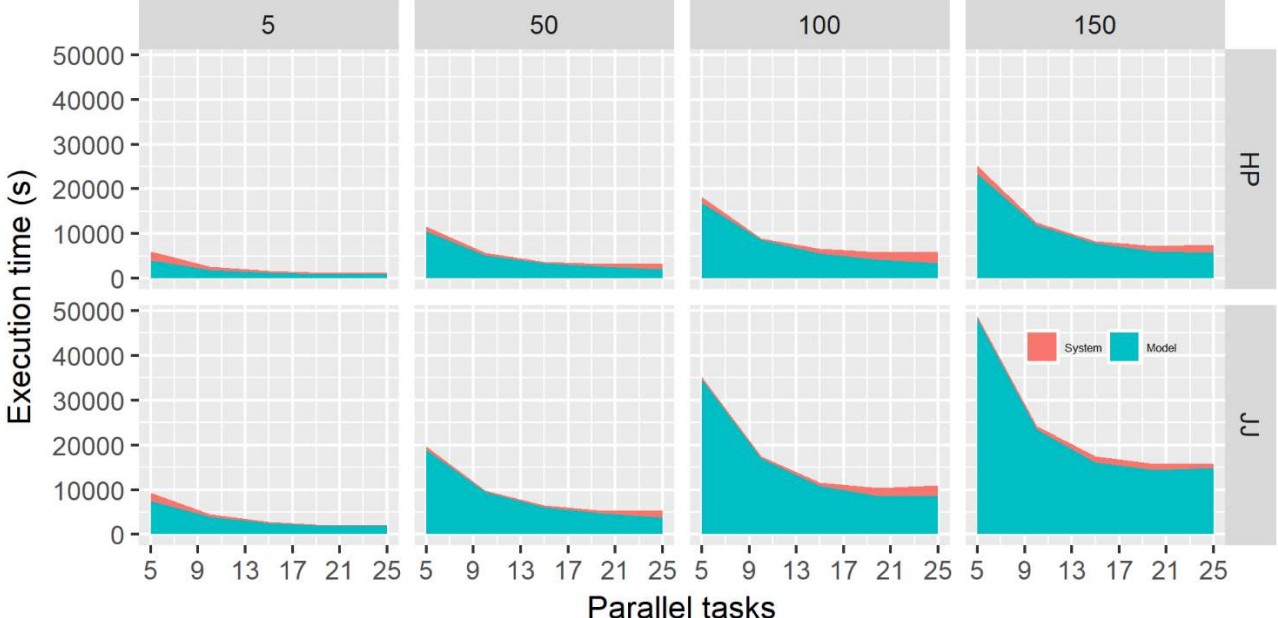

**Figure 9: Job execution times on a Spark cluster versus the number of parallel tasks for the eight synthetic hydrological models.**

## 4.2 Possible usage and guidance

As a two-level parallelization framework, GP-SWAT can be used to reduce the run time of single or iterative model simulations. For modeling routines involving iterative model simulations, such as model calibration, sensitivity and uncertainty analysis, and beneficial management practice (BMP) optimization, users can incorporate GP-SWAT into the

350 corresponding algorithms/tools with proper adaptation to enhance the model simulation performance. Moreover, when GP-SWAT is operating in the decomposed mode (i.e., subbasin-level parallelization), it can be very computationally frugal in some cases. For example, when GP-SWAT is used for BMP optimization or subbasin-scale model calibration, the changes in model input pertain to only a small portion of the model components (e.g., a subbasin or reservoir). In such cases, only the changed and downstream subbasins and reservoirs require new computations during the iterative simulation process, thus

greatly reducing the computational complexity through the reuse of the model results for the unchanged subbasins. Compared with the parallelization of iterative model simulations, the parallelization of a single model is of less interest because most model applications involve a large number of simulations. However, we have found that it is essential to accelerate single model simulation when using hydrological models to support applications such as emergency decision-making or flood forecasting in a product environment. Because of its ability to accelerate a single model, we believe that GP-

SWAT can be integrated into decision-making and flood forecasting systems to offer quasi-real-time support for decision-making and flood forecasting.

GP-SWAT can be run in either the decomposed mode or the integrated mode. As indicated by the experimental results, the operation mode can exert a great influence on the performance of GP-SWAT. Compared with the integrated mode, the decomposed mode requires more computation time to perform task management and model result transfer; thus, it may not be suitable for lightweight models. On the other hand, the decomposed mode can greatly alleviate the I/O bottleneck for complex models, thereby reducing the computation time. In such a case, the extra time required for task management and model result transfer is negligible. Therefore, we suggest running GP-SWAT in the integrated mode for lightweight models and running it in the decomposed mode otherwise. There is one exception to this suggestion. When conducting iterative simulations of lightweight models, GP-SWAT should run in the decomposed mode if the changes are restricted to a small portion of the downstream subbasins. In addition, the performance of GP-SWAT can be affected by the characteristics of the watershed route network. In general, the decomposed mode is less suitable for watersheds that are thin and long than for watersheds with short paths and many branches.

## 4.3 Advantages and limitations

GP-SWAT represents the first attempt to accelerate the simulation of a highly computationally intense SWAT model through two-level parallelization using the graph-parallel Pregel algorithm. As a two-level model parallelization tool, it can be used to accelerate single or iterative model simulations, endowing it with a range of possible applications and great flexibility in maximizing performance. In addition, the applicability of a model parallelization tool is also influenced by its running environment. As an open-source tool implemented in Java and Scala, GP-SWAT can be run not only on a commodity computer with a Linux, Unix or Windows OS but also on a Spark cluster with a large number of computational nodes. Moreover, the Spark framework is so universal in the IT industry that many cloud providers currently offer convenient on-demand managed Spark clusters (e.g., Google Dataproc, Amazon Web Services EMR, and Azure HDInsight) with out-of-the-box support for Spark-based applications. Thus, users can easily adapt GP-SWAT to run in these environments, thereby avoiding the technical and financial burdens encountered when building an in-house Spark cluster.

Moreover, the Spark-based implementation of GP-SWAT lends it many advantages that other distributed computation frameworks (e.g., MPI) may not have. First, Spark provides many high-level functionalities, such as distributed task dispatching, scheduling, failover management and load balancing, which can greatly reduce the burden on programmers. For example, by means of the failover management functionality, Spark-based programs can gracefully handle partial failures (e.g., partial breakdown of the computational nodes), as the Spark framework can automatically detect failed tasks and then reschedule these tasks on healthy computational nodes. In MPI-based applications, it may be necessary to explicitly manage these issues. Second, the Spark framework is a higher-level parallel computing framework. The SWAT model and Spark need to be coupled only loosely to achieve model parallelization. In contrast, to achieve model parallelization with MPI, model reconstruction is usually required.

GP-SWAT uses the NFS protocol to exchange data among different computational nodes. While NFS is easy to implement, some problems may be encountered in the case of a large-scale computational cluster. Because NFS employs a master-slave

architecture, performance bottlenecks may arise when a large number of nodes are simultaneously attempting to read from or write to the master. Another drawback of GP-SWAT is that it requires additional I/O operations when working in the decomposed mode (i.e., subbasin-level parallelization). For example, the basin-level inputs are replicated to the subbasin models, and point-source files (which contain redundancies with the standard SWAT outputs) are generated and read by the subbasin models. However, this issue can be partially addressed by optimizing the I/O module of SWAT to reduce the redundant output.

## 5 Summary and future work

In this study, we developed a two-level (i.e., model-level and subbasin-level) parallel simulation framework for the SWAT model, called GP-SWAT, based on the graph-parallel Pregel algorithm. The efficiency of GP-SWAT was evaluated through two sets of experiments. Experiment set one was conducted to illustrate that GP-SWAT can be used to perform subbasin-level model parallelization using a multicore computer running the Windows OS. The results show that GP-SWAT can accelerate the single model simulation process, especially for complex models. In experiment set two, GP-SWAT was assessed for iterative model runs. For each experiment in this set, subbasin- and watershed-level parallelization schemes were employed to execute 1000 model simulations in the test environment with one to five parallel tasks implemented at each computational node. Our experimental results show that GP-SWAT can achieve a remarkable performance improvement over traditional SWAT models by leveraging the computational power of a Spark cluster. In addition to performance improvement, GP-SWAT also has some other notable features.

(1) It can be employed to perform both individual and iterative model parallelization, endowing it with a range of possible applications and great flexibility in maximizing performance through the selection of a suitable parallelization mode (at the model level or the subbasin level).

(2) It is a flexible and scalable tool that can run in diverse environments, ranging from a commodity computer with a Microsoft Windows, Mac, or Linux OS to a Spark cluster consisting of a large number of computational nodes, either deployed in-house or provided by a third-party cloud service provider.

Further work should also be conducted to improve the performance and extend the usability of our proposed method. In this study, data exchange was achieved through the NFS protocol, which can present an I/O bottleneck when a large number of computational nodes are involved. To scale GP-SWAT to a larger computational cluster, a distributed storage system, such as the Hadoop Distributed File System (HDFS) or Redis, should be used to address this potential issue. In addition, the application of the proposed scheme to other environmental models, such as the Hydrologic Simulation Program-FORTRAN (HSPF) model, will be necessary to demonstrate the flexibility and universality of our proposed method.

## Code availability

The model code (GP-SWAT v1.0) is released by MIT under license. Source codes, manual and synthetic models used in this study (https://doi.org/10.5281/zenodo.4270676) are available on the Zenodo repository.

## Author contribution

Dejian Zhang: Methodology, Writing - original draft, Software, Writing - review & editing, Funding acquisition. Bingqing Lin: Methodology, Writing - original draft, Software, Writing - review & editing. Qiaoying Lin: Conceptualization, Funding acquisition, Supervision, Writing - review & editing. Jiefeng Wu: Writing - original draft, Software, Writing - review & editing, Validation.

## Competing interests

The authors declare that they have no conflict of interest.

## Acknowledgments

This work was financially supported by the Natural Science Foundation of Fujian Province [grant number 2020J01779], the Science and Technology Project of Xiamen [grant number 3502Z20183056], and the Science and Technology Climbing Program of Xiamen University of Technology [grant number XPDKT19014].

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
