# Peer review of "GP-SWAT (v1.0): A two-level graph-based parallel simulation tool for the SWAT model"

_Geoscientific Model Development, 2020_

## Author Comment (AC1)

Dear Reviewers,

Thank you very much for your time and help in reviewing our manuscript. We also want to thank you for your positive and constructive comments, which were very valuable and helpful for improving the quality of our manuscript. We have studied the comments carefully and have made corresponding revisions. Our responses to your comments are listed as follows. The comments are shown in blue, our responses are shown in black, and the revisions are shown in red.

Sincerely,
Qiaoying Lin, Ph.D.

Department of Resources and Environmental Sciences, Quanzhou Normal University, Donghai Street 398, Quanzhou, Fujian 362000, China

Reviewer #1:
This manuscript presents a two-layer graph-based parallel simulation framework for the SWAT model, which can provide valuable reference for watershed management. I suggest the manuscript can be accepted after major revision.
Reply: We truly appreciate your positive comments and effort in reviewing our manuscript.

1. The specific results of the parallel-computing performance should be presented in the abstract.
Reply: We have included the performance measures under different parallelization strategies in the abstract.

"…The single model parallelization results showed that GP-SWAT can achieve a 1.8-5.8-times speedup. For multiple simulations with subbasin-level parallelization, GP-SWAT yielded a remarkable speedup of 6.62-27.03 times. In both cases, the speedup ratios increased with increasing model complexity…"

2. Line 95: It's better to state why this research wants to propose a new parallelization scheme?
Reply: We have added the purpose of this study accordingly.

"…The objective of this study is to create a simulation-accelerated tool for the SWAT model by adopting both watershed-level and subbasin-level parallelization, without the model reconstruction to achieve data communication among model components that current solutions (e.g., MPI) usually require. We hope that this tool will help IT practitioners or modelers improve model performance without requiring specific domain knowledge of the hydrological model…"

3. Section 2.1 should be integrated to the introduction part, which can help to figure out what is the missing part of existing researches.
Reply: This has been done accordingly.

4. Fig. 3: it is not convinced for most reader to read the codes. Some diagrams are needed to express the same meanings.
Reply: We added a diagram alongside the source code to illustrate the components and processes used

to achieve model parallelization with the Pregel algorithm.

[Figure]

Figure 3: Schematic diagram of model parallelization with the Pregel algorithm (a) and code snippet of the driver program (b) (note that imported packages and some signatures have been removed for simplicity).

5. Fig 5: How can the actual speedup ratios larger than the theoretical ones?

Reply: The reason the actual speedup values surpass the theoretical values is because the speedup is achieved not only by simulating the model components in parallel but also by alleviating the model's IO burden. For example, the execution time of the undivided model of JJ4 is 425.02 s; nevertheless, the execution time for its subbasin model is only 2.35 s. In other words, even executing the subbasin model of JJ4 in sequence can still lead to a reduction of 192.37 s (425.02-2.35*99). This is because the original SWAT model reads in all model inputs at a time in the initial stages of model execution, which can easily cause I/O saturation and thus prolong the model execution time. The theoretical speedup ratios were calculated against the execution time of the original model. In contrast to the theoretical speedup ratio calculation, the actual speedup ratios were calculated by considering both performance gains from subbasin-level parallelization and IO alleviation. We added the above explanation to the results section.

"…It is noted that the actual speedup values surpass the theoretical values for HP3, HP4, JJ3 and JJ4. The reason of this phenomenon is that the speedup is achieved not only by simulating the model components in parallel but also by alleviating the model's IO burden. For example, the execution time of the undivided model of JJ4 is 425.02 s, but the execution time of its subbasin model is only 2.35 s. In other words, even if the subbasin models of JJ4 are executed in sequence, this still can lead to a reduction of 192.37 s (425.02-2.35*99). This is because the original SWAT model reads all model inputs at once in the initial stages of model execution, which can easily cause I/O saturation and thus prolong the model execution time. The theoretical speedup ratios were calculated against the execution time of the original model. In contrast to the theoretical

speedup ratio calculation, the actual speedup ratios were calculated by considering both performance gains from subbasin-level parallelization and IO alleviation…"

6. Line 230: some details of the study areas should be added.
Reply: More information pertaining to these two study areas was added to the revised manuscript.

"…In this study, synthetic hydrological models of the Harp Lake and Jinjiang watersheds were used to evaluate the performance of GP-SWAT. The Harp Lake and Jinjiang watersheds have distinct characteristics in terms of spatial extents and stream network complexities and were thus used as representatives of simple and complex watersheds, respectively, to demonstrate how watershed characteristics can affect the GP-SWAT performance. The Harp Lake catchment, which covers a drainage area of 5.20 km2, is located within the Boreal Shield ecozone in south-central Ontario, Canada. It has six streams that drain directly into Harp Lake. The land uses include mixed deciduous and coniferous forest (>75% of the total land area), lakes (about 15%), and wetlands and ponds (about 7%). The annual mean temperature ranges from 3.5 °C to 6.5 °C, with an average of 4.9 °C. The mean January and July air temperatures are -10.5 °C and 18.4 °C, respectively. Jinjiang catchment is located in south-eastern Fujian Province of China. It has a drainage area of 5629 km2, which is occupied predominantly by mountains and rangelands. There are two major river branches within the Jinjiang River, i.e., the east branch and the west branch, which merge into the main stream 2.5 km upstream of the Shilong gauge station. With a typical humid sub-tropical climate, the area has an annual mean temperature of 20 °C and average annual precipitation of 1686 mm…"

[Figure]

Figure 5: Watershed delineations for the Harp Lake (a) and Jinjiang (b) watersheds.

7. Line 235: it seems that the computation amount is not very large in the case study. I wonder how the proposed parallel computing system performs for the applications with many different amount of simulation units.
Reply: To evaluate the performance of GP-SWAT under various IO burdens, 8 synthetic models were built based on the Harp Lake and Jinjiang hydrological models. The synthetic models based on the Jinjiang hydrological model (JJ) with 5, 50, 100 and 150 HRUs per subbasin are denoted by JJ1, JJ2, JJ3 and JJ4, respectively. The synthetic models based on the Harp Lake hydrological model (HP) with

5, 50, 100 and 150 HRUs per subbasin are denoted by HP1, HP2, HP3 and HP4, respectively. As all JJ-based synthetic models contain 99 subbasins, the total HRUs of the JJ1, JJ2, JJ3 and JJ4 models are 495, 4950, 9900 and 14850, respectively. With 38 subbasins, the total HRUs of HP1, HP2, HP3 and HP4 are 190, 1900, 3800 and 5700, respectively. Currently, the maximum number of HRUs allowed in SWAT is 20000; thus, we believe that these synthetic models can reasonably serve as test models with respect to IO burden testing. We are now conducting a follow-up study of GP-SWAT, in which we inspect how the network complexities and the manner of their arrangement can affect the performance of GP-SWAT. We would like to present your concern in an ongoing study. We sincerely hope for your understanding. More text describing the synthetic models has been added in the case study section.
* * *
Reviewer #2:
General comments:

This manuscript proposed a unified parallelization strategy for both watershed-level and subbasin-level parallelization and developed the GP-SWAT accordingly, which is valuable and useful for both developers and users in the scientific community. Except for the specific comments posted by Anonymous Referee #1 (https://gmd.copernicus.org/preprints/gmd-2020-429#RC1), I have some other specific comments.

Reply: Thank you for your constructive comments. We have accounted for all of your comments in the revised manuscript.

**Specific comments:**
1. The phase "two-layer graph-based parallelization" is ambiguous and inexact. (1) Do you mean "layer" equals "level" since you use "model-level" (or "watershed-level", please unify the terms) and "subbasin-level" in the manuscript? (2) The "graph-based" is the parallelization strategy for "subbasin-level" not for "model-level", since model runs are independent to each other and only their outputs are concerned.

Reply: We have unified these phases to eliminate possible confusion. GP-SWAT works at both the watershed level and subbasin level. Figures 1a, b and c show how the graph can be represented for parallelization of a single model at the subbasin level, iterative model simulations at the subbasin level, and iterative model simulations at the watershed level, respectively. In the simplest case, i.e., the parallelization of a single model at the subbasin level, each node in the graph represents a subbasin model, and each edge denotes an upstream-downstream flow drainage relationship. In the case of iterative model simulations at the subbasin level, each node in the graph represents a subbasin model of a specific simulation. In other words, some of the nodes in the graph may represent the same subbasin model with different parameters with respect to certain simulations. To form an integrated graph, a virtual vertex (denoted by -1) is added and connected to the outlet vertex (figure 1b). In the case of iterative model simulations at the watershed level, each node in a graph represents a watershed model, and a virtual node is added and connected to a watershed node to form an integrated property graph. GP-SWAT can choose to execute subbasin models and watershed models or do nothing according to the types of nodes.

[Figure]

Figure 1: Directed acyclic graphs represent watershed route information (a) and iterative simulations at the subbasin level (b) and model level (c).

2. In the introduction, the first paragraph stated that both the single model and numerous models require prohibitive execution time. However, the first three of the four methods introduced in the second paragraph are used for model-level applications. In my view, it is more clear to introduce the

methods for alleviating computational burden for the model level applications and the single model, separately. And then both focus on parallel computing.

Reply: Thank you for your suggestion. We have revised the introduction section according to your and reviewer one's comments. In the introduction section, we first introduce the significance of this study, briefly review model-level, submodel-level, spatial-decomposition parallelization methods and the effects of parallel computing techniques and resources on model performance, and finally, outline the research content and goals.

3. In the sentence around No 65, it is more precisely to cite Liu et al. (2016) and/or Zhu et al. (2019) rather than Liu et al. (2014).

Reply: This has been revised accordingly.

4. In the sentence around No 80, "However, these methods have two major limitations: … complex computational facilities that may not be readily available…" should be reconsidered. In fact, many parallel computing models (e.g., MPI) can also be running on the personal computer and obtain a good speedup ratio.

Reply: We agree. We have removed this statement from our revised manuscript.

5. This manuscript focuses on the parallelization of both model-level and subbasin-level but missed existing similar methods such as Zhu et al. (2019).

Reply: We carefully studied your work, which presented an integrated model parallelization paradigm to maximize model performance through basic-unit-, subbasin- and model-level parallelization, at the very beginning of planning this study. This study was inspired by the work of Wang et al. (2013), Yalew et al. (2013), Liu et al. (2014, 2016), Zhu et al. (2019) and other studies mentioned in the introduction section. We adopted only parts of the parallelization methods proposed in these studies because we intended to develop a generic parallelization method that can be used to speed up model simulation without model reconstruction and to maximize the potential of applying our proposed framework to other environmental models (although it was only tested with the SWAT model in this study). For modelers who want to maximize their model performance, we highly encourage them to incorporate these parallelization methods as well.

References:

Wang, H., Fu, X., Wang, Y., and Wang, G.: A High-performance temporal-spatial discretization method for the parallel computing of river basins, Computers & Geosciences, 58, 62–68, https://doi.org/10.1016/j.cageo.2013.04.026, 2013.

Yalew, S., van Griensven, A., Ray, N., Kokoszkiewicz, L., and Betrie, G. D.: Distributed computation of large scale SWAT models on the Grid, Environ. Modell. Softw., 41, 223-230, https://doi.org/10.1016/j.envsoft.2012.08.002, 2013.

Liu, J., Zhu, A. X., Liu, Y., Zhu, T., and Qin, C. Z.: A layered approach to parallel computing for spatially distributed hydrological modeling, Environ. Modell. Softw., 51, 221-227, https://doi.org/10.1016/j.envsoft.2013.10.005, 2014.

Liu, J., Zhu, A.-X., Qin, C.-Z., Wu, H., and Jiang, J.: A two-level parallelization method for distributed hydrological models, Environmental Modelling & Software, 80, 175–184, https://doi.org/10.1016/j.envsoft.2016.02.032, 2016.

Zhu, L.-J., Liu, J., Qin, C.-Z., and Zhu, A.-X.: A modular and parallelized watershed modeling framework, Environmental Modelling & Software, 122, 104526, https://doi.org/10.1016/j.envsoft.2019.104526, 2019.

6. Overall, the introduction failed to raise the scientific issue clearly and precisely, that is, there is no unified parallelization strategy for both watershed-level and subbasin-level parallelization that do not need to reconstruct source code of hydrologic model to handle data communication among subbasins explicitly (e.g., using MPI). If this is correct, the title may also be changed accordingly.

Reply: Thank you for your suggestion. We revised the introduction section in our revised manuscript according to your suggestion.

7. What is the phrase "the current computation step" mean in the sentence around No 140? To my understanding, the proposed method is different from the spatial-temporal discretization proposed by Wang et al. (2013). For example, in Fig 4, Subbasin 1 and Subbasin 2 are executed for the entire simulation period (e.g., 5 years) first, then Subbasin 3 begins to run, and so on. This may lead to a poor load balance, and hence a low speed up ratio, especially for a single model run. Is this right?

Reply: The computation step in this sentence is a "Superstep" of the Pregel algorithm. We replaced this phase with "the current Superstep" in accordance with the Pregel algorithm. To avoid model reconstruction, the parallelization method that we proposed was not extended to the temporal discretization method originally proposed by Wang et al. (2013). Indeed, the actual speedup ratio that can be achieved is largely dependent on the structure of the stream network. However, the spatial discretization method can effectively reduce the IO burden (mainly concentrated in the model initial period) of the original model by distrusting IO over the entire simulation period, thus further improving the overall performance of GP-SWAT.

8. In Section 4.1, all the results are compared through speed-up ratios, you should also give the actual execution times. I want to know the performance of the subbasin-level parallelization for a single model simulation compared with the original SWAT model.

Reply: The actual execution times associated with different parallelization strategies and the original model were added to Figures 7 and 9 and analyzed in the Results section.

"…Figure 7 shows the job execution times of these synthetic models versus the parallel tasks performed on a Windows server. The job execution time includes the times associated with job initiation, task orchestration and model result transfer (system time) and the time pertaining to model execution (model time). The model execution times were calculated by summing the total execution time for each subbasin model, and the system execution times were calculated by subtracting the model execution times from the job execution times. As shown in Figure 7, the system execution times were stable for both the HP-based and JJ-based models with different HRUs…"

[Figure]

Figure 7: Job execution times on a Windows server versus the number of parallel tasks for the eight synthetic hydrological models.

"…Figure 9 shows the execution times against the number of parallel tasks. In general, the fraction of system time decreases with increasing parallel tasks and increases again when the number of parallel tasks reaches 4 and/or 5. We believe that larger system times with smaller parallel tasks occur because the model execution overheads are relatively small, while in cases with large parallel tasks, the conflict of demanding resources certainly leads to a longer time required for performing task dispatch and securing computational resources…"

[Figure]

Figure 9: Job execution times on a Spark cluster versus the number of parallel tasks for the eight synthetic hydrological models.

9. Why use two study areas? From my perspective, the two case studies have no significant difference. We need more information about the study areas.

Reply: The intention of using two study areas in this study was to demonstrate how stream network complexities can affect GP-SWAT performance. The synthetic models (i.e., HP1, HP2, HP3 and HP4) based on the Harp Lake hydrological model have a relatively simple stream network, while the synthetic models generated from the Jinjiang hydrological model (i.e., JJ1, JJ2, JJ3 and JJ4) were used to represent complex models with respect to stream network characteristics. More information pertaining to these two study areas was added to the revised manuscript.

"…In this study, synthetic hydrological models of the Harp Lake and Jinjiang watersheds were used to evaluate the performance of GP-SWAT. The Harp Lake and Jinjiang watersheds have distinct characteristics in terms of spatial extents and stream network complexities and were thus used as representatives of simple and complex watersheds, respectively, to demonstrate how watershed characteristics can affect the GP-SWAT performance. The Harp Lake catchment, which is a covered a drainage area of 5.20 km$^2$, is located within the Boreal Shield ecozone in south-central Ontario, Canada. It has six streams that drain directly into Harp Lake. The land uses include mixed deciduous and coniferous forest (>75% of the total land area), lakes (about 15%), and wetlands and ponds (about 7%). The annual mean temperature ranges from 3.5 °C to 6.5 °C, with an average of 4.9 °C. The mean January and July air temperatures are -10.5 °C and 18.4 °C, respectively. Jinjiang catchment is located in south-eastern Fujian Province of China. It has a drainage area of 5629 km$^2$, which is occupied predominantly by mountains and rangelands. There are two major river branches within the Jinjiang River, i.e., the east branch and the west branch, which merge into the main stream 2.5 km upstream of the Shilong gauge station. With a typical humid sub-tropical climate, the area has an annual mean temperature of 20 °C and average annual

precipitation of 1686 mm…"

10. In sentences around No 235, it is weird that the numbers of HRUs per subbasin can be set the same.

Reply: We used a tool that was developed in a previous study to generate eight synthetic models that were used to evaluate the performance of GP-SWAT. This tool allows users to create synthetic models with any number of HRUs per subbasin. These synthetic models were also included in the Zenodo repository (https://doi.org/10.5281/zenodo.4447969) along with the source codes and the tutorial of GP-SWAT. We have added more information on how these synthetic models were created in the revised manuscript.

"…These synthetic models were created with a tool implemented in Java Program Language (available at https://github.com/djzhang80/models-and-tools/createproject.jar). This tool allows users to create synthetic models with any desired HRUs per subbasin for performance evaluation but not to create meaningful hydrological models…"

**Technical corrections:**
1. Did you mean that the "Spark-SWAT" is the alias of the "GP-SWAT"?

Reply: We unified the name of our developed tool in the revised manuscript.

2. In the code, all file paths are specific to the author's computer. This is not suitable for code distribution. Even so, the modification of these paths should be clarified in the tutorial.

Reply: The paths of the dependent libraries and the input files in the source codes have been replaced with path variables instead, and information pertaining to these paths is included in the tutorial.

---

## Author Response (AR2)

Dear Reviewer,

Thank you very much for reviewing our manuscript. We also want to thank you for your insightful comments, which were very valuable and helpful in improving the quality of our manuscript. We have studied the comments carefully and have made corresponding revisions. Our responses to your comments are listed as follows. The comments are shown in blue, and our responses are shown in black.

Sincerely,
Qiaoying Lin, Ph.D.

Department of Resources and Environmental Sciences, Quanzhou Normal University, Donghai Street 398, Quanzhou, Fujian 362000, China

Reviewer #2:
General comments:

The revised manuscript is greatly improved, especially the introduction section. Most of the comments have been well addressed. However, I still have some specific comments.
The basic idea of this manuscript is to alleviate the development burden of hydrological modelers to achieve high-performance watershed modeling without reconstruction of model code, which is novel and clearly stated. The implementation based on the SWAT model, i.e., GP-SWAT, must be helpful for the scientific community. Overall, I am glad to suggest an acceptance for publication after a minor revision.

Reply: Thank you for your constructive comments. We have addressed all of your comments in the revised manuscript.

Specific comments:
1. In Line 53-54, the author introduced three types of parallelization strategies, such as model-level, submodel-level, and spatial-decomposition. But, in my view, the author has confused the spatial-decomposition method with the submodel-level, i.e., Line 64-79 should be the spatial-decomposition method, or more precisely, the spatial(-temporal) decomposition method, and Line 80-90 should be the submodel-level method. I mean, the so-called submodel level is a special case derived from the spatial(-temporal) decomposition method. In such a case, each submodel is a full model executed on one part of the watershed (i.e., subbasin). Besides, each parallelization type should have a short and precise definition. Please consider my suggestion.

Reply: Thank you very much for your suggestion. We have revised the introduction section thoroughly to avoid possible confusion regarding the parallelization strategies introduced. In the introduction section, we briefly review the broad model parallelization method and then narrow it to a special model parallelization method, i.e., the spatial-decomposition method. When reviewing the spatial-decomposition method, we first introduced spatial-decomposition implemented through model reconstruction, which is a possible limitation. We then described another spatial-decomposition method

performed without model reconstruction, which has advantages (reducing the modeler's workload and providing an alternative to cooperate with recent advanced information technologies and resources). We finally outlined the goal and scheme of the research presented in this paper. The introduction section is written as follows:

[revised manuscript text omitted]

2. The title used "a…simulation framework", but the introduction only listed some parallelization strategies (or named parallelization schemes). I would suggest introducing existing hydrological modeling frameworks based on parallel computing and raise their weakness. I think that will be the answer to the second comment of #referee 1 (Line 95: It's better to state why this research wants to propose a new parallelization scheme?). Also, in the main text, the author used "a two-level parallelization scheme", why not "framework", and what is the difference?

Reply: Thank you for your valuable suggestion. We have unified these phases to eliminate any possible confusion. The proposed method that is able to perform two-level model parallelization in this study is referred to as the scheme, and the software we developed accordingly is referred to as the two-level model parallelization tool. The introduction section is concisely revised to focus on the spatial-decomposition method. When the limitations of spatial-decomposition through model reconstruction are outlined, the advantages and disadvantages of using existing hydrological modeling frameworks such as the Open MPI and the OpenMP application programming interface (API) are briefly introduced as follows:

"However, this parallelization method is relatively complex, as it requires a throughout model reconstruction to enable the parallel simulation of model components, to perform the communication among components that is necessary for integrating the model results, and to deal with issues such as failover and load balance. As a result, a steep learning curve is expected for modelers who are unfamiliar with the model source codes. Although there are some parallel computation frameworks available that can facilitate this method, e.g., the Open MPI and the OpenMP application programming interface (API), it is still a very tedious and time-consuming process."

3. The authors claimed that "indeed, the actual speedup ratio that can be achieved is largely dependent on the structure of the stream network." and "The intention of using two study areas in this study was to demonstrate how stream network complexities can affect GP-SWAT performance". Although the revised manuscript added some more descriptions of the two study areas, I cannot find the quantitative or qualitative analysis of the different stream networks' structures and the consequent result differences. So, I may suggest only retain the Jinjiang study area. Or, if the author can give a calculation method of theoretical speedup ratio considering the structure of stream networks and the available computing resources, that will be much valuable to adopt the two distinct study areas.

Reply: We agree. Only the Jinjiang study area is retained in the revised manuscript. Statements pertaining to the Harp Lake catchment and stream network structure have been removed throughout the manuscript. We are planning a new study to investigate how the structure of the stream network and the organization of the directed acyclic graph can affect the performance of GP-SWAT. We hope this issue can be well addressed in our future research. The calculation of the theoretical speedup is included and is defined as follows:

$$Speed_{ref} = Sub_{num}/\sum_{i=1}^{n} Ceil(Count_i/PT_{num}),\qquad\qquad (2)$$

where $sub_{num}$ is the number of subbasins of the hydrological model under test, n is the total number of supersteps, $i$ denotes the $i$-th superstep, $Count_i$ is the number of subbasin models simulated in the $i$-th superstep, $PT_{num}$ is the number of parallel tasks performed at an executor, and $Ceil$ is a function that returns the smallest integer value that is greater than or equal to a predetermined parameter value.